# Protecting local water quality has global benefits

John A. Downing [1,2,3✉], Stephen Polasky [4,5], Sheila M. Olmstead [6,7] & Stephen C. Newbold[8]

Surface water is among Earth's most important resources. Yet, benefit–cost studies often report that the costs of water quality protection exceed its benefits. One possible reason for this seeming paradox is that often only a narrow range of local water quality benefits are considered. In particular, the climate damages from water pollution have rarely been quantified. Recent advances in global water science allow the computation of the global methane emission from lakes caused by human nutrient enrichment (eutrophication). Here, we estimate the present value of the global social cost of eutrophication-driven methane emissions from lakes between 2015 and 2050 to be $7.5–$81 trillion (2015 $US), and in a case-study for one well-studied lake (Lake Erie) we find the global value of avoiding eutrophication exceeds local values of either beach use or sport fishing by 10-fold.

[1] Minnesota Sea Grant, University of Minnesota, Duluth, MN 55812-1198, USA. [2] Department of Biology, University of Minnesota Duluth, Duluth, MN 55812, USA. [3] Large Lakes Observatory, Research Lab Building, Duluth, MN 55812, USA. [4] Ecology, Department of Evolution and Behavior, University of Minnesota, St. Paul, MN 55108, USA. [5] Department of Applied Economics, University of Minnesota, St. Paul, MN 55108, USA. [6] Lyndon B. Johnson School of Public Affairs, The University of Texas at Austin, Austin, TX 78713-8925, USA. [7] Resources for the Future, Washington, DC 20036, USA. [8] Department of Economics, University of Wyoming, Laramie, WY 82071, USA. ✉email: downing@umn.edu

Clean freshwater is a key strategic resource[1]. Water pollution has been at or near the top of the list of U.S. environmental concerns for the past 30 years[2], and many believe that surface water supplies are now at least somewhat dangerous[3]. Nonetheless, economic studies of water quality regulations in the U.S. often report estimates of benefits smaller than costs[3–5]. One reason for low estimates of economic benefits is that many ecosystem services supported by clean water are poorly understood and not included in estimates of benefits[6]. Studies of the economic benefits of air pollution use well-established links from emissions to concentrations, exposure, health outcomes, and finally to monetary value. For water quality, similar well-established links of water quality to economic benefits are more limited. We combine recent limnological and biogeochemical models with advances in integrated economic assessment to calculate an important aspect of the value of surface water quality not previously monetized: reducing nutrient pollution in lakes and reservoirs reduces eutrophication, which in turn leads to lower methane ($CH_4$) emissions that impact climate.

Eutrophication is a problem of great economic importance[7]. Aquatic scientists have recently estimated greenhouse gas (GHG) emissions from lakes and reservoirs, and the growth in GHG emissions from increased eutrophication[8] associated with rising phosphorus (P) and nitrogen (N) pollution expected over the next century[9,10]. These peer-reviewed analyses are based on the most geographically extensive data set collected to date and cover 8000 lakes from a broad diversity of climates and geographic regions, including all continents and many observations from the tropics[10]. Eutrophication of inland waters, driving emissions of $CH_4$, is forecast to increase up to nearly 5-fold over the next century (Table 1 in ref. [9]) due to population growth, agricultural expansion, warming of surface waters, increased storminess, and expansion of waters in places susceptible to eutrophication.

Prior work monetizes the damages from nitrous oxide ($N_2O$) associated with anthropogenic active N release to the environment in the United States[11] and the European Union[12]. Some other studies have estimated the aggregate global value of ecosystem services from lakes and rivers for food and water provision, waste treatment, and recreation[13,14]. No prior studies have monetized global damages from eutrophication-related emissions of $CH_4$, however, even though $CH_4$ constitutes 75% of the atmospheric impact from lakes and reservoirs and now contributes annual emissions of 0.55–1.0 Pg $CH_4$ yr$^{-1}$ [10], with an influence on climate change comparable to about 20% of the current emissions from fossil fuel combustion. Here we monetize the global social cost of current and future $CH_4$ emissions due to lake eutrophication.

Eutrophication is expected to increase by 20–100% by 2050 and up to 120–390% by 2100 under business-as-usual climate and population projections. By 2100, $CH_4$ emissions from lakes and reservoirs could have an impact on climate change equivalent to about 38–53% of current fossil fuel emissions. If GHG mitigation reduces emissions from fossil fuel use, eutrophication's share of GHG emissions will rise even further. If eutrophication increases at these projected rates, future $CH_4$ emissions from lakes and reservoirs are likely to counterbalance the totality of marine carbon burial or all terrestrial carbon burial in the global carbon budget[9].

In this work, we calculate the global climate damages from $CH_4$ emissions and the benefits of avoided damages from preventing projected increases in rates of eutrophication from 2015 to 2050. To help put our estimates into context, we also implement a case-study of local vs global damages for Lake Erie, one of the five Great Lakes of North America[7].

## Results and discussion

**Global value of controlling eutrophication**. The substantial emissions from lakes and reservoirs and the potential for increased emissions suggest that there is considerable value in improving water quality in lakes and reservoirs and in preventing further deterioration. We calculated the global climate damages from $CH_4$ emissions and the avoided damages from preventing increased emissions from 2015 to 2050 using well-accepted integrated assessment models (IAMs) (see "Methods"). Because GHGs rapidly become well mixed in the atmosphere, the global social costs of GHG emissions do not depend on where they are emitted. Because GHGs can persist for many years in the atmosphere, the effect of emissions of today will be felt for many years in the future, which means that the rate used to discount future economic damages to the present exerts a strong influence on the social cost of GHG (SC-GHG) estimates. Following the U.S. Government Interagency Working Group (IWG), we report all results using three discount rates: 2.5%, 3%, and 5% yr$^{-1}$.

The estimated present value of the global climate change costs of $CH_4$ emissions from lakes and reservoirs for 2015–2050 range from $7.5 to 81 trillion (2015$; top half of Table 1). Low-end estimates assume a high discount rate (5% yr$^{-1}$), low current emissions (4.8 Pg $CO_2$-eq yr$^{-1}$), and no emission growth. High-end estimates assume a low discount rate (2.5% yr$^{-1}$), high

### Table 1 Present value (PV) of global social costs of $CH_4$ emissions from lakes and reservoirs, 2015–2050 (billion 2015 US$).

| | PV Low constant[a] (1) | PV High constant[b] (2) | PV Low rising[c] (3) | PV High rising[d] (4) |
|---|---|---|---|---|
| **SC-$CH_4$ method[e]** | | | | |
| Discount rate = 5% | 7496 | 14,056 | 8159 | 19,217 |
| Discount rate = 3% | 21,545 | 40,396 | 23,643 | 57,599 |
| Discount rate = 2.5% | 30,144 | 56,520 | 33,120 | 81,015 |
| **SC-$CO_2$ × $CO_2$-e method[f]** | | | | |
| Discount rate = 5% | 5419 | 10,162 | 5881 | 13,655 |
| Discount rate = 3% | 23,017 | 43,157 | 25,147 | 60,158 |
| Discount rate = 2.5% | 36,110 | 67,706 | 39,493 | 94,873 |

[a]Low constant estimates assume low current emissions from lakes (4.8 Pg $CO_2$-eq yr$^{-1}$), and no change in emissions over time.
[b]High constant estimates assume high current emissions from lakes (8.4 Pg $CO_2$-eq yr$^{-1}$), which stay constant over time.
[c]Low rising estimates assume low current emissions, but assume emissions growth of 20%, 2015–2050.
[d]High rising estimates assume high current emissions, as well as high growth over time (100%, 2015–2050).
[e]SC-$CH_4$ method uses estimates of the social costs of $CO_2$, $CH_4$, and $N_2O$ adapted from published sources[16,29].
[f]SC-$CO_2$ × $CO_2$-e method converts $CH_4$ to $CO_2$-equivalents and uses estimates of the social cost of carbon dioxide[15].

current emissions (8.4 Pg $CO_2$-eq $yr^{-1}$), and high growth in emissions from lakes (100%). It will not be possible to avoid all emissions from lakes and reservoirs, but with concerted effort it may be possible to prevent increased emissions. The present value of avoided damages from holding emissions constant at current levels rather than increasing by 20–100% by 2050 from increasing eutrophication is $0.66–24 trillion (2015$).

Although it has been noted that it might result in under-estimation, especially when assuming a high discount rate[15], an alternative approach to estimating the climate change damages from non-$CO_2$ GHGs involves first converting the emissions to $CO_2$-equivalents ($CO_2$-eq)[16] and then multiplying these by the social cost of carbon dioxide (SC-$CO_2$)[15]. This approach is less accurate than direct application of the social cost of $CH_4$ (SC-$CH_4$)[15], but it has been frequently used in previous studies. To facilitate comparison to other estimates of climate damages in the literature, we also used the $CO_2$-e × SC-$CO_2$ approach with otherwise equivalent assumptions to value eutrophication emissions. Results using this approach are reported in the bottom half of Table 1. The cost of $CH_4$ emissions from lakes and reservoirs from 2015 to 2050 is estimated to be $5.4–95 trillion (2015$), and the associated avoided damages from keeping emissions constant are $0.46–27 trillion (2015$).

These estimates consider only the cost of $CH_4$ emissions, but lakes and reservoirs also emit $CO_2$ and $N_2O$. Adding current $CO_2$ and $N_2O$ emission estimates[10], the SC-GHG emissions increases by 27–51% above those for $CH_4$ alone. Although mounting evidence suggests poor water quality also influences emissions of $CO_2$ and $N_2O$, global analyses of future scenarios for altered emissions of $CO_2$ and $N_2O$ from lakes have not yet been published, so we do not monetize these damages. Nevertheless, even our partial estimates suggest that reducing eutrophication is an important means of avoiding climate change damages with a large benefit when measured in monetary terms.

**Comparison to other economic damages from water pollution.** How do these estimated global climate damages from eutrophica-tion compare to the local and regional benefits of water pollution control typically included in assessments of the benefits and costs of water pollution policies? To help put our results in context, we consider the case of Lake Erie, where eutrophication and associated harmful algal blooms (HABs), primarily due to excess P from agricultural sources, have caused considerable economic damage since the mid-1990s[7]. Local values of eutrophication abatement vary among lakes, but Lake Erie is a salient example because reliable estimates of local value are available, and Lake Erie's GHG emis-sions were included in the global emission analysis[9,10] that we used to compute our global estimates presented in Table 1. Recent work using a stated preference survey of Ohio residents estimates that a 40% reduction in total P loading to the western Lake Erie basin from the Maumee River watershed would lead to a $4.0–6.0 million annual welfare gain to Ohio recreational anglers[17,18]. Assuming constant annual benefits from 2015 to 2050 and using a 3% $yr^{-1}$ discount rate, this amounts to a present value of $0.087–0.12 billion in total recreational fishing benefits.

Applying our methods to this case, a 40% reduction in total P loading to Lake Erie would yield a 0.079 Tg $yr^{-1}$ reduction in $CH_4$ emissions (2.7 Tg $CO_2$-eq $yr^{-1}$). If the P-loading reduction began in 2015 and was maintained through 2050, we estimate that the resulting water quality improvement would generate present value economic benefits (avoided climate damages) of $3.1 billion using the SC-$CH_4$ ($3.3 billion using $CO_2$-e × SC-$CO_2$) and a 3% $yr^{-1}$ discount rate (Table 2). Thus, the global climate benefits of achieving the targeted 40% reduction in P

**Table 2 Present value (PV) of avoided global social costs of $CH_4$ emissions, 2015–2050 (billion 2015 US$), from a 40% reduction in total P loading in the western Lake Erie basin[a].**

|  | PV |
|---|---|
| SC-$CH_4$ method[b] |  |
| Discount rate = 5% | 1.08 |
| Discount rate = 3% | 3.11 |
| Discount rate = 2.5% | 4.36 |
| SC-$CO_2$ × $CO_2$-e method[c] |  |
| Discount rate = 5% | 0.78 |
| Discount rate = 3% | 3.33 |
| Discount rate = 2.5% | 5.22 |

[a]A 40% reduction in total P loading would yield a 2.696 Tg $CO_2$-eq $yr^{-1}$ $CO_2$-eq flux of (100 year) reduction in $CH_4$ emissions (0.07929 Tg $CH_4$ $yr^{-1}$).
[b]SC-$CH_4$ method uses estimates of the social costs of $CO_2$, $CH_4$, and $N_2O$ adapted from published sources[16,29].
[c]SC-$CO_2$ × $CO_2$-e method converts $CH_4$ to $CO_2$-equivalents and uses estimates of the social cost of carbon dioxide[15].

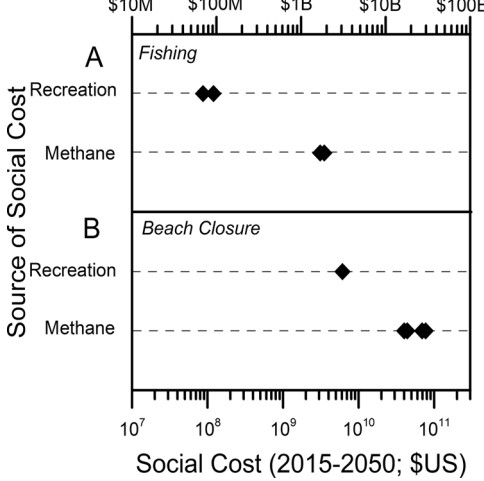

**Fig. 1 Comparison of the recreational vs. climate implications of eutrophication. A** The welfare gain, 2015–2050, from a 40% reduction in phosphorus (P) loading to western Lake Erie reducing the frequency and extent of harmful algal blooms (HABs). The range of economic impact on recreational angling was estimated from the annual welfare gain[17] assuming constant annual benefits and a 3% $yr^{-1}$ discount rate. The welfare gain from this same total P loading to Lake Erie was estimated from the corresponding reduction in $CH_4$ emissions (and $CO_2$-equivalent emissions) through 2050, using estimates and methods reported in Table 2. **B** The welfare cost of seasonal Lake Erie HABs sufficient to close beaches, 2015–2050. Benefit transfer work[20] estimates the 95% confidence interval of daily recreational losses from the closure of all 67 Lake Erie beaches in Ohio and Michigan. We aggregate to seasonal (115 day)[39] HABs occurring annually, 2015–2050, using a 3% $yr^{-1}$ discount rate. Methane cost estimates are derived from methane emissions under nutrient concentrations that would lead to closure of all of these beaches due to high chlorophyll from HABs as well as from chlorophyll levels that would lead to moderate risk of adverse health effects from beach use.

loading are well over an order of magnitude larger than the estimated recreational benefits to Ohio anglers (Fig. 1).

Published estimates suggest that the 40% reduction in total P loading to Lake Erie that we model here could be achieved with a fertilizer tax or a tax-and-rebate policy with rebates funding agricultural best management practices at an annual cost to

taxpayers of about $16–17 million[19]. Note that these cost estimates are conservative, as they do not include yield losses or other agricultural compliance costs[19]. These annual costs would exceed the estimated annual recreational fishing benefits of the policy goal[18] but are still smaller than the climate benefits.

Economists have also used benefit transfer techniques to extrapolate from individual estimates of the value of water quality changes for a specific location to estimates for an entire region. For example, recent work[20] using a function transfer approach estimates that the closure of all 67 Lake Erie beaches in Ohio and Michigan due to a large HAB in Lake Erie would generate daily recreational losses of $2.39 million (95% confidence interval $1.81–3.11 million). Assuming an extreme case that the HAB season lasts continuously for 115 days[20], this implies an annual welfare loss of about $280 million. If a severe HAB that closed all 67 Lake Erie beaches in the two states occurs annually from 2015 to 2050 and annual damages are constant, the present value of total damages, derived from the definition of the present value of a constant stream of benefits, using a 3% $yr^{-1}$ discount rate, would be about $6.1 billion using the central estimate of the cost of beach closure[20], or a range of $4.4–7.7 billion, using their 95% confidence interval[20].

The $CH_4$ emissions from a HAB event in Lake Erie large enough to close all 67 beaches in Ohio and Michigan would depend on the severity of the triggering water quality impairment. We use two approaches to make a comparable estimate of $CH_4$ emission damages. First, if the chlorophyll *a* concentration exceeds 30 ppb, the risk of Cyanobacteria blooms is 80–100%, gauged by the risk of Cyanobacteria biomass exceeding 50%[21]. This level would exceed statutory thresholds that trigger beach closures or health advisories and would yield an emission increase of 1.0 Tg $CH_4$ $yr^{-1}$ (34 Tg $CO_2$-eq $yr^{-1}$). These emissions would create a present value of damages of $39 billion using the SC-$CH_4$ ($42 billion using $CO_2$-e × SC-$CO_2$) at a 3% $yr^{-1}$ discount rate (Table 3), roughly seven times larger than the estimated recreational damages from a HAB severe enough to close all Lake Erie beaches in Michigan and Ohio for 35 years.

As a second approach to making this comparison, we use the World Health Organization guideline for chlorophyll *a* concentration yielding a moderate probability of adverse health effects in recreational waters (50 ppb)[22]. Because the assumed triggering concentration for beach closures is higher, both the estimated emissions associated with the closure events (1.7 Tg $CH_4$ $yr^{-1}$ or 59 Tg $CO_2$-eq $yr^{-1}$) and the economic damages using a 3% $yr^{-1}$ discount rate ($69 and $73 billion) are higher (Table 3). With this approach, the global climate costs of HABs severe enough to close all MI and OH beaches on Lake Erie from 2015 to 2050 are an order of magnitude larger than the estimated recreational damages from beach closures (Fig. 1).

We cannot say how our $CH_4$ damage estimates would compare with a full estimate of other damages from Lake Erie eutrophication. The literature demonstrates that important water quality benefits are difficult to value[2]. A single-season HAB similar to the 2014 event that resulted in the issuance of a do not drink/do not boil order for the public water system in the City of Toledo created damages of about $1.3 billion, including impacts on property values, water treatment costs, and tourism[23]. Estimates of damages to fishing activity at Lake Erie's Canadian coast are also substantial[24]. An earlier study estimates damages from eutrophication of all U.S. rivers and lakes[25], omitting the climate damage estimates we calculate here; an assessment of the methods used to obtain these estimates is outside the scope of our paper. Notably, recent work links HABs in Gull Lake, Michigan (not far from Lake Erie) with increased likelihood of low birth weight and shorter gestation among infants born to exposed mothers[26].

Given that the full gamut of potential damages is difficult to monetize, a comprehensive estimate of the non-climate damages from eutrophication and HABs—especially if human health impacts are significant—could exceed our damage estimates for $CH_4$ emissions. However, our estimates of the global $CH_4$ emission damages from eutrophication in Lake Erie exceed all published estimates of other damages, to the extent that we can compare them. Smaller lakes than Erie may show even greater differences between global and local values of eutrophication because, on average, people have greater willingness to pay for recreation on large lakes[27], and $CH_4$ emissions per unit area do not vary with lake size[10]. These results suggest that global climate impacts are a substantial omission from benefit–cost assessments of policies targeting eutrophication, in Lake Erie and elsewhere.

**Eutrophication is a local and global problem**. Degraded water quality is often considered a local or regional problem. We show that water quality has important implications for global climate, through emissions of $CH_4$ and other GHGs. These emissions are likely to increase substantially unless action is taken to prevent further eutrophication. The damage from eutrophication-related GHG emissions is likely to be in trillions of dollars, and appears to be far larger than other monetized damages from poor water quality that economists have so far been able to quantify, especially where pollution does not generate severe health damages. Our analysis shows that local water quality protection has global economic implications, and that more effort devoted to

---

**Table 3 Present value (PV) of global social costs of $CH_4$ emissions, 2015–2050 (billion 2015 US$), from a harmful algal bloom sufficient to close all MI and OH beaches on Lake Erie.**

| | PV<br>Closure at 30 ppb chlorophyll *a*[a] | PV<br>Closure at 50 ppb chlorophyll *a*[b] |
|---|---|---|
| SC-$CH_4$ method[c] | | |
| Discount rate = 5% | 13.72 | 23.87 |
| Discount rate = 3% | 39.42 | 68.59 |
| Discount rate = 2.5% | 55.16 | 95.97 |
| SC-$CO_2$ × $CO_2$-e method[d] | | |
| Discount rate = 5% | 9.92 | 17.25 |
| Discount rate = 3% | 42.12 | 73.28 |
| Discount rate = 2.5% | 66.08 | 114.96 |

[a]A 30 ppb chlorophyll *a* concentration represents an 80–100% risk of Cyanobacteria blooms[21]; we associate this with a 1.003926 Tg $yr^{-1}$ increase in $CH_4$ emissions.
[b]The World Health Organization chlorophyll *a* guideline for avoiding moderate probability of adverse health effects in recreational waters is 50 ppb[22]; our estimates suggest an associated increase of 1.746587 Tg $yr^{-1}$ in $CH_4$ emissions.
[c]SC-$CH_4$ method uses estimates of the social costs of $CO_2$, $CH_4$, and $N_2O$ adapted from published sources[16,29].
[d]SC-$CO_2$ × $CO_2$-e method converts $CH_4$ to $CO_2$-equivalents and uses estimates of the social cost of carbon dioxide[15].

understanding the consequences of changes in water quality and valuing the benefits of sustaining or improving water quality is warranted.

## Methods

**Computation of climate damage from methane emissions.** To compute the climate damages of $CH_4$ emissions from lakes, we used estimates of the social costs of carbon dioxide ($CO_2$), $CH_4$, and nitrous oxide ($N_2O$) produced by the U.S. Government IWG on the SC-GHGs[28,29]. The IWG used a common set of input assumptions and three IAMs—DICE[15], FUND[30], and PAGE[31]—to calculate the discounted value of the expected future global economic losses from climate change due to emissions of each GHG between 2015 and 2050.

**Integrated assessment models and their limitations.** DICE, FUND, and PAGE are among the main IAMs used for benefit–cost analysis of climate change policies in the U.S. and elsewhere[32]. These IAMs combine a reduced-form representation of the influence of GHG emissions on global average temperatures with estimates of the economic damages from increasing temperatures over time[16,28]. Well-known limitations of IAMs stem from disagreements about how economic damages from climate change in the far future should be compared to the near-term costs of emissions reductions, and uncertainties about the impacts of changes in the physical climate on economic systems, including but not limited to the risks of catastrophic economic impacts if large increases in global temperatures are reached[33–36].

## Data availability

All data used for estimating global GHG emissions from lakes are available in the FigShare repository[37] at https://doi.org/10.6084/m9.figshare.5220001.

## Code availability

No complex code was used in the creation of this manuscript. However, a spreadsheet of the principal calculations in Tables 1–3 is available[38] at https://doi.org/10.6084/m9.figshare.14265188 (2021).

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

## Acknowledgements

We thank Cathy Kling, Dave Keiser, and Dan Phaneuf for convening a workshop on Integrated Assessment Models and the Social Costs of Water Pollution, under the auspices and support of the Atkinson Center of Cornell University. We also thank Chris Moore of the US Environmental Protection Agency for signaling a key connection among us that greatly assisted this work.

## Author contributions

J.A.D. and S.P. conceived the idea and plan behind the manuscript. S.C.N. calculated the present values of methane emissions and S.M.O. created comparative analyses of local and global costs of eutrophication. J.A.D. was responsible for all limnological aspects of the manuscript. All authors contributed to writing and revising the manuscript.

## Competing interests

The authors declare no competing interests.
