## [Peer Review File · Nature Communications]

REVIEWER COMMENTS

Reviewer #1 (Remarks to the Author):

This is an excellent article that points out another negative consequence of lake eutrophication that extends to the global scale. The unique aspect of the work is it takes recent global work linking eutrophication to methane production, and combines that with methods to estimate economic costs associated with methane release in the atmosphere. The detailed comparisons on the Great Lakes further strengthen the paper as the global estimates have wide ranges, and the methodology in the citations they compare to in the Great Lakes is considerably more detailed than some larger-scale analyses.

I think the largest advance of this paper is that it takes estimating freshwater ecosystem service values another step further. Most papers on this have left big holes in some areas where estimates simply could not be made. This point could be made more strongly; the paper mainly starts discussing the relative costs near the end, and some of the comparisons that were not made here (e.g. property values, cost to clean drinking water, protection of biodiversity).

There also might be some weakness in the global upscaling that needs discussion. The papers with data on the methane efflux rate estimates as a function of lake productivity come from several hundred lakes. Looking into those sources, I could not easily find the spatial distribution of these lakes. I suspect that, as is true of most "global" data sets based on published literature, that tropical and subtropical lakes are underrepresented. This could actually lead to underestimates of methane and nitrous oxide flux for several reasons. 1) Higher temperatures encourage higher metabolic rates, and more anoxia, 2) amictic lakes are more common in tropics, and relatively oligotrophic lakes still can have anoxic hypolimnia. And 3) the longer growing season in the tropics can lead to more production that ultimately fuels methanogenesis or the conditions leading to methane and nitrous oxide production. Further, some of the global benefits are not the same as local. For example, many developed countries rely on fish harvest as a vital source of protein, so fishing could assume a greater importance.

The specific examples from the Great Lakes might be somewhat misleading, as those lakes have a tremendous area. Most lakes are smaller and in developed regions would be expected to generate far more value per unit area of lake via recreation, fishing, and property values (ie. many of these benefits depend on shoreline length, and the length to area ration is greater for smaller lakes.)

It could be worth noting that the global cost of methane production by lakes estimated in this paper exceed \$5 trillion. However the global value of river and lake ecosystems from a few years back was only set at \$ 2.5 trillion (Costanza et al. 2014). You would have to access the supplementary data to discuss the missing cells in their matrix used to calculate total value.

Line 36. The increase in eutrophication probably needs a citation.

Costanza, R., R. de Groot, P. Sutton, S. van der Ploeg, S. J. Anderson, I. Kubiszewski, S. Farber, and R. K. Turner. 2014. Changes in the global value of ecosystem services. *Global Environmental Change* 26:152-158.

Reviewer #2 (Remarks to the Author):

This is an interesting article. It examines the avoided climate change damage costs as a result of lower methane (CH₄) emission levels associated with reduced eutrophication. In doing so, the authors try to identify the relative significance, in economic terms, of this externality associated with

eutrophication. First at a global scale and then illustrated for Lake Erie, one of the 5 Great Lakes in North America. Crucial in the analysis is available information or assumptions about dose-response relationships between phosphorous loadings going into lakes and reservoirs, their impact on water quality and eutrophication levels, the consequent emission of methane into the air and ultimately how this affects the expected climate change related damage costs. Recent studies have shown that increasing eutrophication is expected to increase methane (CH₄) emission levels, and hence increase the impact of climate change. The authors use this existing evidence about the relationship between eutrophication in lakes and reservoirs and methane emission levels to estimate the associated damage costs based on a simplified versions of the climate model MAGICC v5.3 and the model DICE and a global estimate for the social cost of methane. How the integration of the different inputs and outputs between these two models looks like remains a bit unclear. Existing scenarios predicting emission levels of methane from lakes and reservoirs between 2015 and 2050 are used and a present value is calculated of the total methane damage costs. Ensuring that the expected level of eutrophication between 2015 and 2050 is reduced, lowers the emission level of methane and hence avoids the climate-related damage costs. How this can be achieved and at what cost is unclear.

The relative contribution of the economic value of this externality is illustrated for Lake Erie using existing studies of other externalities of eutrophication, mainly related to water-based recreation. The authors show that the avoided damage costs related to methane emissions are orders of magnitude higher than the economic value of other eutrophication related damage costs, and should be accounted for in cost-benefit analyses of water pollution.

The paper's main contribution is that it uses relationships between eutrophication and methane emissions from previous studies and multiplies this with an existing global estimate for the social costs of methane to be able to estimate the avoided damage costs of the climate impacts of methane emissions. It hence monetizes the relationship between eutrophication and methane emissions by examining how the expected increase in eutrophication worldwide contributes to methane emission levels and hence climate damage costs. Very little attention is paid to how reliable these relationships are and hence also how transferable to Lake Erie. Dose-response relationships are borrowed from existing studies and literature. The same goes for the estimation of the global and local costs of the contribution of methane emissions to climate change impacts. The same global relationships are seemingly unconditionally transferred to a specific local lake in North America.

The question is to what extent extra value is added in this study by multiplying in a last step the expected emission level of methane with a global estimate for the social costs of methane emissions. The illustration of the order of magnitude of this externality for Lake Erie is interesting but how representative is it for all lakes and reservoirs around the world? Why was specifically Lake Erie chosen and not for example the whole Great Lakes basin? Would the results look differently if another lake was selected? What if other damage cost categories related to eutrophication are included (water treatment, commercial fishing, impacts on properties along the lakeshore, agriculture)? How would the comparison then look like?

Finally, estimating the dose-response relationships highlighted in this paper are surrounded by uncertainties. The credibility of the economic monetization exercise is not discussed but deserves more attention than currently is the case. The same applies to how the climate model and the economic model are integrated. Integrated assessment models are well-known for having different sources of uncertainty and are better understood using a range of values rather than point estimates. These ranges due to model uncertainty are not mentioned in the text but would be a valuable addition to test the robustness of the estimates. The transferability of rough global estimates to local level also deserves more attention.

In conclusion, I'm in doubt how much of a new contribution this is to the existing literature. The authors base their analysis on existing evidence in the literature. I'm afraid to say that there is in that sense not very much new in the paper, except for the monetization of the avoided damage costs of

climate change if methane emission levels go down. This is based on recent empirical evidence from other studies showing a correlation between eutrophication and methane emission levels. Assuming that the predicted increase in eutrophication will not occur over the period 2015-2050 as the basis for the calculation of the avoided damage costs is perhaps a bit simple. The same applies to the 40% emission reduction of phosphorous in the Great Lakes. No attention is paid to how this can be realized.

Reviewer #3 (Remarks to the Author):

Re: Protecting local water quality has global benefits

Summary: This paper uses an integrated approach to monetize global damages from CH₄-induced eutrophication in lakes and reservoirs. Its findings add to the literature measuring the benefits associated with clean water. A study case for Lake Erie is presented.

Suggestions and comments:

1. Inconsistencies between numbers reported with the main text and tables.

• Page 3, Paragraph 3.

i) "range from \$7.9-75 trillion". However, the low-end number, low current emissions (4.6 Pg CO₂-e y⁻¹), and low emissions growth (20%) reads as \$8.6 (column 3 in table 1). Also, low current emissions are defined as 4.8 Pg CO₂-eq y⁻¹ in table 1 Notes

ii) Last sentence of the paragraph. I couldn't match the dollar estimates with any numbers reported in Table 1 (column 1 or 2).

• Page 5 first paragraph

ii) Applying our methods to this case, a 40% reduction in total P loading to Lake Erie would yield a 0.079 Tg y⁻¹ reduction in CH₄ emissions (2.7 Tg CO₂-eq y⁻¹). Table 2 Notes reads "reduction in CH₄ emissions (0.7929 Tg y⁻¹ CH₄). "

ii) Table 2 reports \$3.5 billion for SC-CO₂ x CO₂-e method and \$3.1 billion for SC-CH₄ method, but the estimates are reversed in the main text.

2. Other suggestions

• Marten et al. (2012) report using SC-CO₂ x CO₂-e methods underestimates the CH₄ emission reductions by 35%. I think it's worth mentioning it in the manuscript.

• "Emissions of CH₄ from increased eutrophication of inland waters are forecast to increase up to 4-fold over the next century due to population growth, agricultural expansion, warming of surface waters, increased storminess, and expansion of waters in places susceptible to eutrophication." Could you provide a reference? According to Bealieu et al. (2019), "enhanced eutrophication of lakes and impoundments will substantially increase CH₄ emissions from these systems (+30–90%) over the next century."

• How many lakes and reservoirs were used in your analysis. This goes beyond the paper's scope, but it would be interesting to see the breakdown of these benefits by regions.

References:

J. J. Bealieu, T. DelSontro, J. A. Downing, Eutrophication will increase methane emissions from lakes and impoundments during the 21st century. *Nature Comm.* 10, 1375 (2019).

A. L. Marten, S. C. Newbold, Estimating the social cost of non-CO₂ GHG emissions: Methane and nitrous oxide. *Energy Policy* 51, 967-972 (2012).

Comment or suggestion	Change made	Location in revised manuscript
Reviewer #1		
Most other papers on estimating freshwater ecosystem service values have left big holes in some areas where estimates simply could not be made. This point should be made more strongly.	We have now strengthened this point in the manuscript	Lines 168-178
Does the global upscaling of methane emissions under-emphasize tropical and subtropical lakes? – this could lead to underestimation because those lakes tend to have high emission rates. Also, local values differ in different places, e.g., fish catch may be more important in some areas than the Great Lakes.	Text is now included indicating that the analysis on which the work is based included many observations from the tropics and is the most geographically diverse analysis on GHG emissions from lakes ever performed and that it included many tropical and subtropical lakes. Further text is now supplied noting that local values of eutrophication abatement vary among lakes but noting that Lake Erie is a salient example because local values have been calculated for this lake and Erie was included in the methane emission data we used.	Lines 42-47 Lines 112-115 Lines 115-118
Great Lakes examples may be misleading because the difference between methane and other values because small lakes have greater ratios of perimeter to area and may generate more, especially in developed regions	With respect to methane emissions, it is now noted that smaller lakes than Erie may show even greater differences between global and local values of eutrophication control because people have greater willingness to pay for recreation on large lakes (Egan et al. 2008) and CH₄ emissions per unit area do not vary with lake size.	Lines 183-186

Costanza et al. 2014 estimates the global value of river and lake ecosystems as \$2.5 T cf. \$5 T here – that should be cited and may take analysis of their SOI.	The Costanza et al. (2014) paper is now cited and placed in context as well as de Groot et al. (2012) that is the source of the data displayed by Costanza et al. (2014).	Lines 52-54
Line 36. Provide citation for the increase in eutrophication.	Reference to Smith et al (2014) now provided as one of many possible references substantiating increased global eutrophication	Line 43 (ref. 8)
Reviewer #2		
Two things need clarification. (1) Clarify how the inputs and outputs from MAGICC v5.3 and DICE models look. (2) Clarify how the reduction of eutrophication between 2015 and 2050 can be achieved and at what cost.	(1) In a new, expanded, Methods section, we have now clarified how the MAGIC and DICE models work. (2) Although this is a bit beyond the scope of our work, we have now included an estimate of the cost of such a eutrophication reduction and note that the cost would be higher than the available local benefits estimates but not as high as the cost of methane emissions.	(1) New Methods section: Lines 201-215 (2) Lines 132-137
Very little attention is paid to the reliability of relationships between eutrophication and methane emissions and between methane emissions and social cost and how transferrable they are to Lake Erie. There is a seeming unconditional transfer of general literature relationships and a specific lake in North America.	Text now included indicating that the analysis on which the work is based included many observations from geographically diverse lakes. Further, Lake Erie was included in the data set and fit predictions well. Text is also now supplied noting that local values of eutrophication abatement vary in diversity and amplitude among lakes but noting that Lake Erie is a salient example because	Lines 44-47 Lines 74-76 Lines 110-118

	local values have been calculated for this lake and Erie was included in the methane emission data we used.	
How representative is the information on Lake Erie to all of the lakes of the world? Why was Lake Erie chosen and not the whole of the Great Lakes? Would the results look different if another lake was selected? What if other damage cost categories like water treatment, commercial fishing, property value, and agriculture. What would the comparison look like then?	Lake Erie was included in the methane emission data set on which our work was based. Also, because of intense interest in this lake, we could make comparisons between local and global benefits/costs because Erie has been studied enough to permit this, unlike the whole of the Great Lakes. This is now explained. Also, the representative nature of Lake Erie is noted because methane emissions per unit area do not vary with lake size. It would be fantastic to be able to compare the global methane costs with complete inventories of social value of lakes but these are extraordinarily rare if they exist at all. Our study is meant to spur such future work. This is an area we would like to examine in follow-up studies.	Please see responses to points 2 and 3 of R1. Lines 38-40 Lines 44-47 Lines 115-118 Lines 183-186
Uncertainties of the dose-response relationships should be discussed. (1) What is the credibility of the economic monetization exercise with respect to climate/economics? Integrated assessments are known to have lots of uncertainties and are better understood using a range of values. This needs to be mentioned/explored and	(1) We have now clarified how the MAGIC and DICE models work as well as the importance of uncertainties. One of the main sources of uncertainty stems from the appropriate discount rate, which is why we present results for the three rates used by the US Federal Government Interagency	(1) Lines 201-215 (2) Lines 112-118

mentioned in the text. (2) The transferability of rough global estimates to the local level also deserves more attention.	Working Group on the Social Cost of Greenhouse Gases. We also note that the global social costs of GHG emissions do not depend on their emission location because they rapidly become well mixed in the atmosphere.	
(1) Need to emphasize what is new since the models are already published. It is simplistic to calculate the avoided costs of eutrophication by assuming that the predicted increase in eutrophication will NOT occur. (2) Also, the 40% reduction in phosphorus in the Great Lakes needs some attention to how that could be realized.	The published models only address global methane emissions not the social value and cost. This aspect has never been calculated before. The main contribution of this paper is in combining synthesizing the work of limnologist with climate and economic models to highlight the value of reducing an important source of greenhouse gas emissions (methane emissions from lakes and reservoirs) not included in prior analyses. The novelty of these results is now emphasized. (2) Although the policies are beyond the scope of this work, we have now given a suggestion about how this could be realized and the relative cost of doing so.	(1) Lines 17-18 Lines 51-60 (2) Lines 132-137
Reviewer #3		
There are inconsistencies between numbers reported in the main text and in the tables.	All data have been recalculated and appropriate updates and changes have been made.	Please see 1a-1c, below
(1a) Page 3 paragraph 3 disagrees with Table 1 (\$7.9T versus \$8.6T)- also low emissions defined as 4.8 Pg in Table 1 notes	Corrections made	Top half of Table 1, Lines 81-84.

(1b) Page 5, 1 st paragraph disagrees with Table 2 notes. 0.079 Tg vs 0.7929 Tg CH4	This typo has been corrected and numbers now match.	Lines 124-125 and Table 2 notes, lines 360-363
(1c) Table 2 values of \$3.5B and \$3.1B are reversed in the text.	This has been corrected and the numbers have been updated	Lines 125-129 and Table 2
Other suggestions		
Marten et al. 2012 report that SC-CO2xCO2-e methods underestimate CH4 by 35% - this is worth mentioning	We have now clarified that this is why we offer calculations based on both approaches. This is now emphasized.	Line 90-93
Provide reference to 4-fold increase. It does not match Beaulieu et al's text.	We had inverted the words "eutrophication" with "methane emissions" in the sentence and underestimated the upper bound (4.91-fold). This error has now been corrected	Lines 47-50
How many lakes were used in the analysis? Another thing beyond the scope of the manuscript – it would be interesting to break down benefits by region	These peer-reviewed analyses are based on the most geographically expansive data set collected to date and cover 8000 lakes from a broad diversity of climates and geographic regions, including all continents and many observations from the tropics. This text is now included as is a reference to the original data source. We now emphasize that the CH4 costs and benefits are global, not regional, because methane is well mixed in the atmosphere.	Lines 44-47 Lines 74-76
OTHER CHANGES MADE	We have updated calculations in the text and Table 3 and redrawn Fig. 1 to agree with data changes.	Lines 162-165 Table 3 Figure 1

REVIEWERS' COMMENTS

Reviewer #1 (Remarks to the Author):

The reviewers have addressed all my points adequately, as well as those of the other reviewers, this is a strong and important paper.

Reviewer #3

No additional comments